# Investigation of Spherical Al_2_O_3_ Magnetic Abrasive Prepared by Novel Method for Finishing of the Inner Surface of Cobalt–Chromium Alloy Cardiovascular Stents Tube

**DOI:** 10.3390/mi14030621

**Published:** 2023-03-08

**Authors:** Guangxin Liu, Yugang Zhao, Zhihao Li, Hanlin Yu, Chen Cao, Jianbing Meng, Haiyun Zhang, Chuang Zhao

**Affiliations:** School of Mechanical Engineering, Shandong University of Technology, Zibo 255049, China

**Keywords:** spherical Al_2_O_3_ magnetic abrasive, cobalt–chromium alloy, cardiovascular stent tubes, plasma molten metal powder, magnetic abrasive finishing

## Abstract

In this investigation, spherical Al_2_O_3_ magnetic abrasive particles (MAPs) were used to polish the inner surface of ultra-fine long cobalt–chromium alloy cardiovascular stent tubes. The magnetic abrasives were prepared by combining plasma molten metal powder and hard abrasives, and the magnetic abrasives prepared by this new method are characterized by high sphericity, narrow particle size distribution range, long life, and good economic value. Firstly, the spherical Al_2_O_3_ magnetic abrasives were prepared by the new method; secondly, the polishing machine for the inner surface of the ultra-fine long cardiovascular stent tubes was developed; finally, the influence laws of spindle speed, magnetic pole speed, MAP filling quantities, the magnetic pole gap on the surface roughness (Ra), and the removal thickness (RT) of tubes were investigated. The results showed that the prepared Al_2_O_3_ magnetic abrasives were spherical in shape, and their superficial layer was tightly bound with Al_2_O_3_ hard abrasives with sharp cutting; the use of spherical Al_2_O_3_ magnetic abrasives could achieve the polishing of the inner surface of ultra-fine cobalt–chromium alloy cardiovascular bracket tubes, and after processing, the inner surface roughness (Ra) of the tubes decreased from 0.337 µm to 0.09 µm and had an RT of 5.106 µm.

## 1. Introduction

Cardiovascular diseases (CVDs), the most prevalent diseases affecting human health, cause more than 18 million deaths annually [1,2,3]. Cardiovascular stenting techniques have become an important tool in the treatment of CVDs because of their unique immediate patency and minimally invasive nature [4,5,6]. The cobalt–chromium (Co–Cr) alloy vascular stent is a medical implant with good mechanical properties, high resistance to wear and tear, and high biocompatibility that is safe and reliable and has obvious clinical benefits [7,8,9]. However, the production process of drawing produces defects such as pits and bumps on the internal surface of cardiovascular stents, which bring high internal surface roughness to the stent tubing, limiting the stable flow of blood, and the detachment of the defective layer can also cause harm to the patient with the implant. Medical implants must have low roughness and, therefore, require the polishing of cobalt–chromium alloy cardiovascular stent tubing [10]. The inner wall of the tubing is polished and the defect layer is removed at the same time. Currently, the polishing of the inner wall of vascular stent tubing is mainly performed chemically [11,12,13]. However, the results of chemical polishing are not satisfactory, with new defects such as pitting and bulging, due to difficulties in the introduction of chemical reagents into ultra-slender vascular stent tubing and the difficulty in controlling them [14]. After chemical polishing, these defects will continue to exist in the processed vascular stent, which is extremely harmful to the majority of patients undergoing vascular stenting procedures, causing blood disorders, vascular inflammation, and secondary blockage of blood vessels. At the same time, improper disposal of the waste solution after chemical polishing can also cause environmental pollution, which is not in line with the concept of green development.

Magnetic abrasive finishing (MAF), a process that employs the magnetic attraction of a composite powder containing hard abrasives and ferromagnetic metals to polish the surface of the workpiece, is adaptive and highly efficient [15,16]. MAF can not only finish flat surfaces [17,18], internal and external cylindrical surfaces [19,20,21], and curved surfaces [22,23], but also deburr a machined surface [24,25]. Currently, MAF has been extensively studied in highly sophisticated industries such as healthcare, communication technology, aerospace, and the military [26,27,28,29,30,31,32]. MAF has now been applied to polish the inner walls of tubes [33,34,35,36]; however, few studies have emerged using magnetic abrasives to polish the inner walls of ultra-fine cobalt–chromium alloy cardiovascular stent tubes. This is due to the irregular shape of existing magnetic abrasives, which not only makes loading into ultra-fine long cardiovascular stent tubing difficult, but also makes the depth of cut of the machined surface inconsistent, limiting the improvement of surface quality. At the same time, the low grinding life of existing magnetic abrasives seriously affects the grinding and polishing efficiency, further limiting the application of magnetic particle grinding in cardiovascular stent tubing polishing. The lack of processing equipment specifically designed for defect removal and inner wall polishing of ultra-long cardiovascular stent tubing is another factor limiting the application of MAF in ultra-long vascular stent tubing polishing. Magnetic abrasives (MAPs), as key components of MAF [15,16], are mainly prepared by the bonding method [37], mechanical mixing method [38,39], atomization fast-setting method [40,41], in situ alloy hardening method [42], hot press sintering method [43], plasma spraying method [44,45,46], and electroless plating method [47]. The bonding method [37] uses a special binder, hard abrasive powder, and iron matrix powder mixed in a certain ratio, and the prepared magnetic abrasive is obtained after curing, crushing, and screening. The low cost and simple process are the characteristics of this preparation method; however, the curing and crushing process leads to irregularity of the prepared magnetic abrasives, which limits its introduction into ultra-fine long cardiovascular support tubes. The addition of hard abrasives that are easily dislodged significantly reduces the grinding life and greatly reduces the efficiency of grinding and polishing the inner wall of the tubing. The mechanical mixing method [38,39], a simple preparation method, results in a magnetic abrasive made by mixing a certain percentage of lipid, hard abrasive powder, and ferromagnetic powder in proportion to each other. Through this, method it is simpler to prepare magnetic abrasives, but in the grinding and polishing process, the ferromagnetic powder and hard abrasives are easily decomposed due to the joint action of centrifugal force and magnetic force, and the ideal polishing effect cannot be achieved. The atomization rapid coagulation method [40,41] results in a magnetic abrasive prepared by using aerosolized powder-making equipment combined with high-speed spraying of hard abrasive, which is rapidly solidified by gas cooling (or water cooling). The magnetic abrasives prepared by this method have outstanding features such as high grinding capacity and long life. However, this method is based on the preparation of powder by the atomization method, and the prepared magnetic abrasives have a normal particle size distribution, which is consistent with the particle size distribution of metal powders prepared by the atomization method [48]. The magnetic abrasives prepared by the atomization fast coagulation method have less than half of the particle size in the effective range, low yield, and serious material waste, and are not economical. The economic value is low when using high-priced diamond or CBN powder as hard abrasives, which is not in line with the green development concept. The in situ alloy hardening method [42] is a magnetic abrasive preparation method that generates a hardened layer on the surface of carbon-based iron powder through a chemical reaction. The magnetic abrasives prepared by this method are spherical in shape, but no sharp cutting edge exists and the abrasive performance is very low, which makes it difficult to remove and polish the defect layer of ultra-long cardiovascular stent tubes. The hot-press sintering method [43] was used to produce magnetic abrasives by crushing a mixture of sintered iron matrix powder and hard abrasives. The magnetic abrasives prepared by this method have the disadvantages of difficult comminution and irregular shape, which make them difficult to introduce into the interior of ultra-fine cobalt–chromium alloy cardiovascular stent tubing. The plasma spraying method [44,45,46] was used to prepare regular spherical magnetic abrasives by using high-temperature melting and spheroidization of the plasma torch. The hard abrasive phase of the magnetic abrasives prepared by this method is severely passivated or even detached due to high temperature, with low bond strength and greatly reduced grinding life. The electroless composite plating method [48] is to add ferromagnetic powder and diamond hard abrasive to the chemical solution, so that diamond abrasive grains and plated metal are co-deposited on the ferromagnetic powder substrate to produce magnetic abrasives. The magnetic abrasives prepared by this method have irregular morphology and low abrasive performance. The magnetic abrasives prepared by the above-described magnetic abrasive preparation methods all show limitations for finishing ultra-long cardiovascular stent tubes.

The magnetic abrasives prepared by the combination method of plasma molten metal powder and hard abrasives have the characteristics of high sphericity, controllable particle size (the particle size of the prepared magnetic abrasives is controlled by adjusting the particle size of the iron matrix in the raw material), long life, and good economic value. The preparation method not only solves the problems of low bonding strength between metal matrix and hard abrasive, short service life, and poor grinding performance, which cannot be overcome in the traditional plasma fused magnetic abrasive preparation method, but also solves the key technical problems of uneven particle size and low yield of magnetic abrasive prepared by the traditional aerosolized magnetic abrasive method, and has the advantages of both, reduces the preparation cost, and improves the preparation quality. It is convenient to introduce ultra-fine and long cobalt–chromium alloy cardiovascular stent tubing for processing, and solves the problem of processing tools used for polishing the inner wall of ultra-fine and ultra-long cobalt–chromium alloy cardiovascular stent tubing. In this study, an ultra-fine and ultra-long cardiovascular stent tubing inner wall polishing machine was also developed to solve the problem of the lack of special processing equipment for defect layer removal and magnetic particle grinding inner wall polishing of ultra-fine and long cardiovascular stent tubing.

In the current work, we have investigated the removal of defective layers from the inner walls of cardiovascular stents using magnetic particle finishing, and the magnetic abrasives used were obtained by atomization [49,50]. As reviewed earlier, the disadvantage of the atomization method when preparing magnetic abrasives from expensive hard abrasives such as CBN and PCD is that there is serious waste, and the available grain size does not exceed 50%. The new method for the preparation of magnetic abrasives proposed in this paper has the characteristics of the atomization method, where hard abrasives are firmly bonded to the iron substrate, and is also able to improve the particle size distribution problem. The focus of this paper is the removal and polishing of defective layers on the inner wall of cardiovascular stent tubing using the new method of magnetic abrasive preparation.

## 2. Preparation Principle and Procedure of Al_2_O_3_ Magnetic Abrasive

The schematic diagram of the equipment for preparing spherical Al_2_O_3_ magnetic abrasive by combining plasma molten metal powder and hard abrasive is shown in Figure 1. The equipment consists of a plasma generator, drum wheel type precision powder mixing device, abrasive spray tray, magnetic abrasive synthesis condensation chamber, gas station and pipeline, power supply and air supply system, vacuum dust removal system, and so on. The principle of preparing spherical Al_2_O_3_ magnetic abrasive is as follows: the spherical iron matrix with particle size within a certain range (e.g., 106~120 μm) is mixed with low-pressure inert gas through the drum wheel type precision powder mixer and delivered to the stable plasma torch for heating, and the spherical iron matrix is heated to form micro-droplets moving down along the plasma torch; meanwhile, the Al_2_O_3_ hard abrasive is mixed with high-pressure plasma torch through the drum wheel type precision powder mixer. At the same time, Al_2_O_3_ hard abrasive is mixed with high-pressure inert gas by the drum wheel type precision powder mixer feeder and accelerated by the Laval type abrasive annular slit spray disc. The Al_2_O_3_ hard abrasive particles reach a very high velocity and shoot into the falling metal micro-droplets like bullets, which only overcome the surface tension of the metal droplets to enter the inside instead of breaking the metal droplets, thus avoiding the generation of even smaller metal droplets and ensuring the prepared magnetic abrasive with narrow particle size distribution range. Finally, the Al_2_O_3_ hard abrasive particles are bonded to the superficial layer of metal particles by rapid condensation (condensation speed up to 10^5^ K/s) to form a high-performance spherical Al_2_O_3_ magnetic abrasive.

The steps of preparing Al_2_O_3_ spherical magnetic abrasive by combining plasma molten metal powder and jet hard powder are as follows: make preparations before magnetic abrasive preparation, place the raw materials in the drum and wheel precision powder mixer, turn on the power of vacuum dust removal system for dust extraction, turn on the power supply and water supply system for power supply, cooling water, and cooling gas, and establish a stable plasma torch for magnetic abrasive preparation. After the preparation, turn off the power supply and air supply system; when the temperature of the abrasive collector is lowered to the same temperature as the room temperature, take off the abrasive collector and obtain the magnetic abrasive, unbound metal powder, and Al_2_O_3_ hard abrasive after sieving, and store them in an airtight container after separation; finally, turn off the power supply of the vacuum and dust removal system.

Table 1 shows the process parameters of the spherical Al_2_O_3_ magnetic abrasive prepared by the combination of plasma molten metal powder and hard abrasives, Figure 2 shows the spray tray structure parameters diagram, and Figure 3 shows the SEM image of the spherical Al_2_O_3_ magnetic abrasive prepared by the new method introduced in this paper. From the figure, it can be seen that the spherical Al_2_O_3_ magnetic abrasive prepared under the process parameters listed in Table 1 has the remarkable advantages of spherical appearance, dense bonding of hard abrasives to the superficial layer of the iron matrix, sharp and exposed cutting edges, and firm bonding, etc. These advantages indicate that the magnetic abrasive has a high grinding capacity. The above advantages of the prepared Al_2_O_3_ magnetic abrasive are due to the high-speed flight of the Al_2_O_3_ hard abrasive during the preparation of the magnetic abrasive, which does not pass through the high-temperature region (over 2730℃ high temperature region), and the very short heating time, so that the cutting edge of the Al_2_O_3_ hard abrasive after the formation of the magnetic abrasive is not blunted by the heat and maintains its original sharpness. The high-speed Al_2_O_3_ hard abrasive only overcomes the surface tension of the metal microdroplets and enters the inside of the microdroplets without breaking the metal microdroplets, thus avoiding the formation of small-size magnetic abrasives and, thus, the size of the prepared magnetic abrasives is effectively controlled.

## 3. Experiment

### 3.1. Processing Equipment and Principles

Figure 4 shows the self-developed polishing machine for magnetic particle finishing processing on the inner wall surface of ultra-slender vascular stent tubes, and the performance parameters of this equipment are shown in Table 2. The polishing machine is composed of a numerical control system developed based on the windows system and a mechanical part, where the mechanical part includes a pedestal, synchronous belt, AC servo motor, magnetic device, and other components. Figure 5 shows a principle diagram of magnetic particle finishing on the inner wall of ultra-slender cardiovascular stent tubes. The working principle of this polishing machine is that the servo motors at both ends of the polishing machine drive the tube rotation, while the magnetic device, through the magnetization of the internal magnetic abrasive to form a magnetic abrasive brush on the inner wall of the tube, produces a small pressure, and the magnetic abrasive brush inside the tube is attracted by the magnetic pole to perform reciprocal movement in the process. In the process of movement, from the servo motors at both ends with the same speed but opposite directions, the magnetic pole attracts the magnetic abrasices. In this process, the tube rotation direction remains unchanged and the magnetic pole device moves reciprocally, which makes the single grain hard abrasive form spatial spiral lines crossing each other on the inner surface of the tube, and the defective layer on the inner wall of the tube will be squeezed by the magnetic abrasive and micro removal, so as to remove and polish the defective layer on the inner surface of the tube. The magnetic pole used has a rectangular groove, and the direction is perpendicular to the tube to improve the polishing efficiency.

The steps of magnetic abrasive finishing of ultra-long cardiovascular support tubes are: (1) fill the tube with magnetic abrasive and cutting fluid proportionally and seal both ends; (2) clamp the tube through the precision chuck to ensure no loosening; (3) adjust the tensioning device for tightening; (4) prepare the CNC program for polishing; (5) after polishing is completed, remove the seal, rinse the tube with anhydrous ethanol, and then air-dry and store in a dry place.

### 3.2. Force Analysis

As shown in Figure 6, a single magnetic abrasive is subjected to two forces inside the cardiovascular support tube: *F_x_* along the isomagnetic potential and *F_y_* along the magnetic force line, and their combined force is *F_m_*, pointing towards the inner wall surface, where *F_m_* causes the magnetic abrasive to make a small indentation depth h against the inner wall surface. The magnetic poles remain in a non-contact mode with the magnetic abrasive inside the tube throughout the process.
(1)$$F_{x}=\left(\frac{4\pi d^{3}}{3}\right)\chi H\frac{\partial H}{\partial x}$$
(2)$$F_{y}=\left(\frac{4\pi d^{3}}{3}\right)\chi H\frac{\partial H}{\partial y}$$
where: *χ*—the magnetization rate of spherical Al_2_O_3_ magnetic abrasives;

*d*—the diameter of the spherical Al_2_O_3_ magnetic abrasive (mm);

*H*—the magnetic field strength at the location being processed (A/m);

$\frac{\partial H}{\partial x}$—the rate of change of the magnetic field along the direction of the magnetic lines of force;

$\frac{\partial H}{\partial y}$—the rate of change of the magnetic field along the magnetic isotope.

### 3.3. Experimental Materials

The ultra-slender cobalt–chromium alloy cardiovascular stent tube samples used for the test were produced by the drawing process, with lengths of 1.8–2.0 m, an outer diameter of 1.8 mm, and an inner diameter of 1.6 mm, shown as Figure 7. The elemental content of the material is listed in Table 3, and the mechanical properties are listed in Table 4.

### 3.4. Finishing Experiments

The effects of spindle speed, magnetic pole speed, MAP filling quantities, magnetic pole gap on the surface roughness (Ra), and removal thickness (RT) of the tube inner wall were investigated. The effective length of each machined tube specimen was 0.5 m. The inside wall roughness (Ra) of the machined tube specimens was measured at 5 mm intervals for each process parameter using DSX1000 Digital Microscopes (OLYMPUS, Tokyo, Japan). The sampling length L was 0.25 mm, and the evaluation length Ln was 5 L. Five measurements were taken to eliminate measurement errors. The tube wall removal thickness (RT) was measured by cutting, inlaying, polishing, and etching under DSX1000 Digital Microscopes, and each section was measured 5 times to eliminate measurement errors, and the average value was calculated as the final value. The experimental parameters shown as Table 5.

## 4. Results and Discussions

### 4.1. The Effect of Spindle Speed on Ra and RT of Tube Inner Wall

The fixed process parameters are magnetic pole speed: 50 mm/min, MAP filling quantities: 0.15 g, and magnetic pole gap: 0.5 mm. The effect of spindle speed on the Ra and RT values of the inner wall of the tube is shown in Figure 8. In the experimental parameters, the Ra value showed a decreasing trend when the spindle speed increased from 100 rpm to 300 rpm, the lowest Ra value was 0.9 μm at 300 rpm, and the Ra value continued to increase to 263 μm as the spindle speed increased beyond 300 rpm, which occurred at 900 rpm. This was due to the fact that once the spindle speed was not at the optimal speed, the magnetic abrasive brush inside the tube was disturbed and the machining accuracy was reduced. The highest value of RT for the inner wall was 5.186 μm at 100 rpm, which tended to decrease as the spindle speed continued to increase. This is due to the fact that the magnetic abrasive brush inside the tube is affected by the spindle speed, and the lower the spindle speed is in the test range, the more stable the magnetic abrasive brush is, which makes the material removal rate increase.

### 4.2. The Effect of Magnetic Pole Speed on Ra and RT of Tube Inner Wall

The fixed process parameters are spindle speed: 300 rpm, MAP filling quantities: 0.15 g, and magnetic pole gap: 0.5 mm. Figure 9 is a graph of the effect of magnetic pole speed on Ra and RT values. It can be seen that Ra and RT values show different trends with increasing pole speed; Ra value increases with increasing magnetic pole speed, while RT value decreases with increasing magnetic pole speed. The extreme values of both occur at a magnetic pole speed of 50 mm/min. This is due to the fact that during machining, when other process parameters are certain, the smaller the magnetic pole speed is, the denser the “spacing” of the formed trajectory and the denser the trajectory lines on the unit machining area are, and the cumulative number of machining per unit machining surface increases, thus improving the machining efficiency and machining accuracy. In contrast, the larger the speed of magnetic pole movement, the larger the “spacing” of the formed trajectory is, the sparser the trajectory lines per unit machining area are, and the cumulative number of machining per unit machining area decreases, thus reducing the machining efficiency and machining accuracy.

### 4.3. The Effect of MAP Filling Quantities on Ra and RT of Tube Inner Wall

The fixed process parameters are spindle speed: 300 rpm, magnetic pole speed: 50 mm/min, and magnetic pole gap: 0.5 mm. Figure 10 shows the graph of the effect of MAP loading quantities on Ra and RT values of the tube inner wall. It can be seen from the graph that with the increase in MAP filling quantities, the Ra value shows a trend of first a sharp decline, then a slow rise, and then a decline. The lowest value of Ra corresponds to the MAP filling quantities of 0.15 g. With the increase in MAP filling quantities of the inner wall, the RT value shows a rising trend that is first sharp and then slow. The MAPs filling quantities of 0.25 g can reach the maximum RT value of 5.2 μm. This is because when the MAP filling quantities in the tube are too large, the thickness of the magnetic abrasive brush becomes large, so the elasticity of the magnetic abrasive brush becomes poor, and MAPs can not flip the cycle, reducing the service life and processing efficiency. When the MAPs filling quantities are small, only a magnetic abrasive brush with low finishing efficiency can be formed, which does not have high finishing performance, while the MAPs filling quantities being moderate results in high finishing performance. The moderate MAP filling quantities have high processing performance, which can remove the defective layer of the inner wall of the tube and polish effectively. From the experimental results, it can be found that the right MAP filling quantities can reach high processing efficiency, while too much MAP filling will cause waste and too little MAP filling will make the processing efficiency lower.

### 4.4. The Effect of Magnetic Pole Gap on Ra and RT of Tube Inner Wall

The fixed process parameters are spindle speed: 300 rpm, magnetic pole speed: 50 mm/min, and MAP filling quantities: 0.15 g. The effect of the magnetic pole gap on the Ra and RT values of the inner wall of the tube is shown in Figure 11. It can be seen from the figure that as the magnetic pole gap increases, the Ra value increases and the RT value decreases. This is due to the fact that smaller pole distance has denser magnetic lines of force and higher magnetic induction strength, which attracts magnetic abrasives more strongly, resulting in harder magnetic abrasive brushes and higher cutting forces, leading to reduced inner wall surface roughness and increased material removal. As the magnetic pole distance continues to increase, the magnetic lines of force diverge and the stiffness of the magnetic abrasive brush decreases, preventing effective material removal and surface roughness reduction.

Figure 12 shows the SEM images after a processing time of 4 h with spherical Al_2_O_3_ magnetic abrasive, where the magnetic abrasive filling is 0.15 g, the spindle speed is 300 r/min, the magnetic abrasive particle size range is 125~150 μm, and slotted magnetic poles are used. After processing with the above parameters, the surface roughness of the inner wall of the tube was reduced from 0.337 μm to Ra = 0.09 μm. Figure 11 shows that the defect layer on the inner wall of the tube was effectively removed after processing, defects such as pits and bumps disappeared, and the surface became smooth. This indicates that the spherical Al_2_O_3_ magnetic abrasive prepared by a new method combining plasma molten metal powder and hard abrasives can remove the defects on the inner wall of cardiovascular stent tubes due to the manufacturing process with excellent finishing effect.

## 5. Conclusions

In this investigation, spherical Al_2_O_3_ magnetic abrasives were prepared by a combined plasma molten metal powder and hard abrasives method, which has potential finishing performance and is expected to overcome the problem of removing and polishing the defect layer on the inner wall of ultra-slender cardiovascular stent tubing. The test results obtained by grinding and polishing the inner surface of cobalt–chromium alloy cardiovascular stent tubing with it led to the following conclusions.

(1) A new method, a combination of plasma molten metal powder and hard abrasives, was used to successfully prepare Al_2_O_3_ magnetic abrasives with high sphericity, controllable particle size, long life and good economic value that are well suited for the removal and polishing of defect layers on the inner surface of ultra-fine and ultra-long cobalt–chromium alloy cardiovascular stents.

(2) CNC processing equipment specifically applied to magnetic abrasives for removing defect layers and polishing the inner wall of ultra-fine and long cardiovascular stent tubing was developed.

(3) This paper uses spherical Al_2_O_3_ magnetic abrasive to finish the inner wall of ultra-fine long cobalt–chromium alloy cardiovascular stent tubing, which can remove surface defects such as bumps and pits generated during the production process.

(4) The effects of spindle speed, magnetic pole speed, MAP filling quantities, and magnetic pole gap on the Ra and RT values of the tube inner wall surface were investigated by single-factor testing, resulting in a reduction in surface roughness (RA) from 0.337 µm to 0.09 µm and RT of 5.106 µm.

## Figures and Tables

**Figure 1 micromachines-14-00621-f001:**
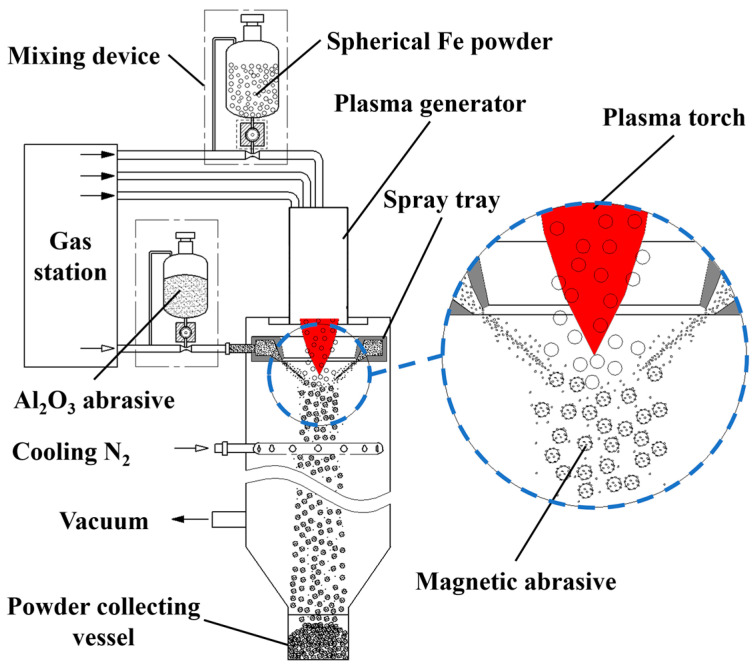
Schematic diagram of combining plasma molten metal powder with sprayed abrasive powder.

**Figure 2 micromachines-14-00621-f002:**
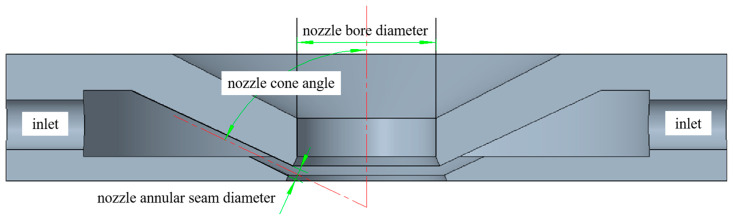
Spray tray structure parameters diagram.

**Figure 3 micromachines-14-00621-f003:**
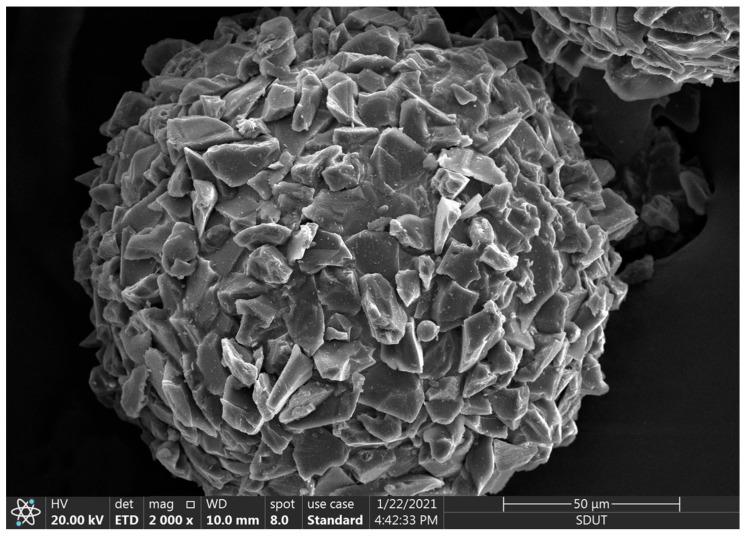
SEM image of spherical Al_2_O_3_ magnetic abrasive.

**Figure 4 micromachines-14-00621-f004:**
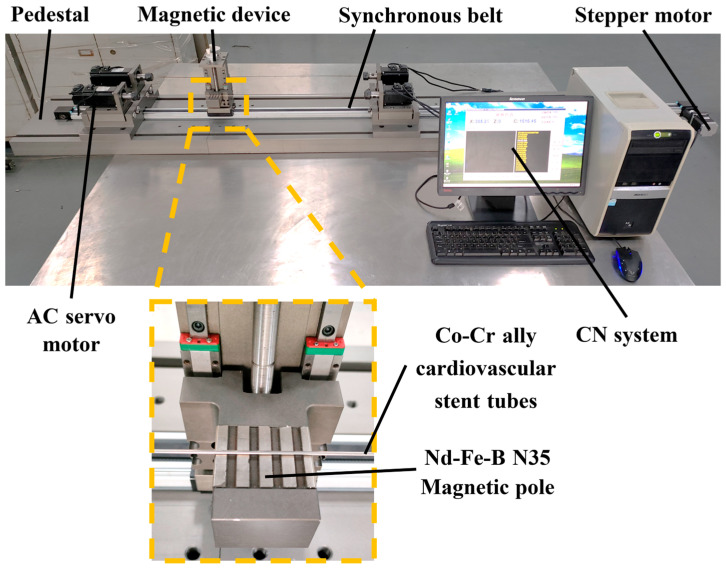
Ultra-slender cardiovascular stent tube inner wall magnetic particle finishing machine.

**Figure 5 micromachines-14-00621-f005:**
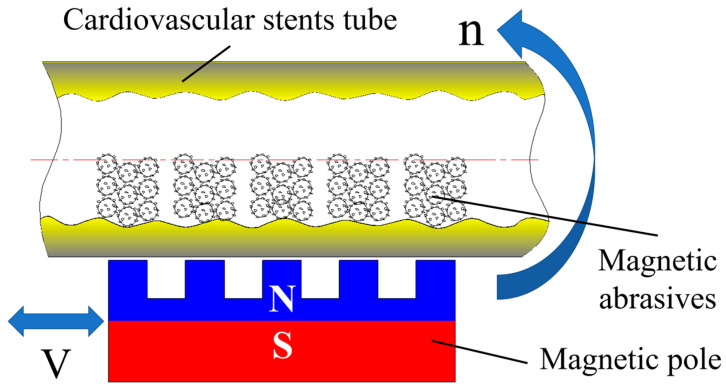
Principle diagram of magnetic particle finishing on the inner wall of ultra-slender cardiovascular stents tube.

**Figure 6 micromachines-14-00621-f006:**
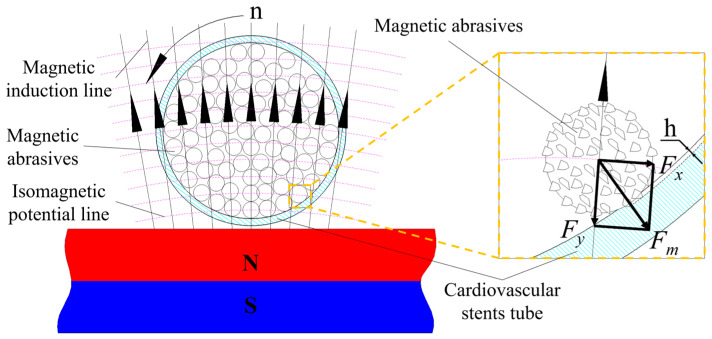
Force diagram of a single magnetic abrasive in a cardiovascular stents tube.

**Figure 7 micromachines-14-00621-f007:**
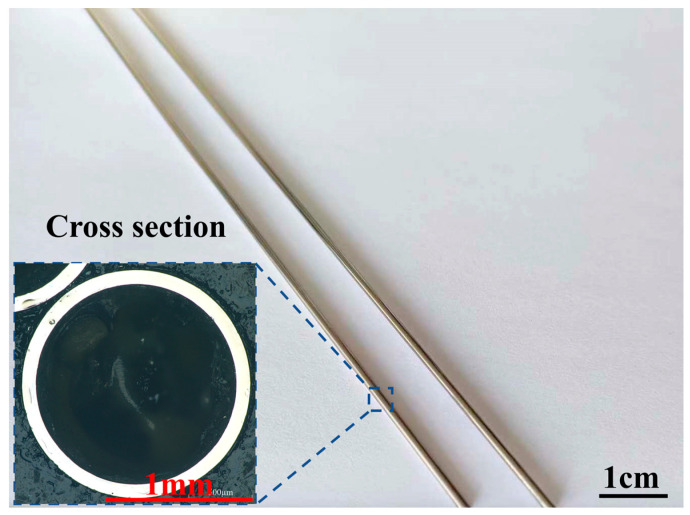
Ultra-slender cobalt–chromium alloy cardiovascular stent tubes.

**Figure 8 micromachines-14-00621-f008:**
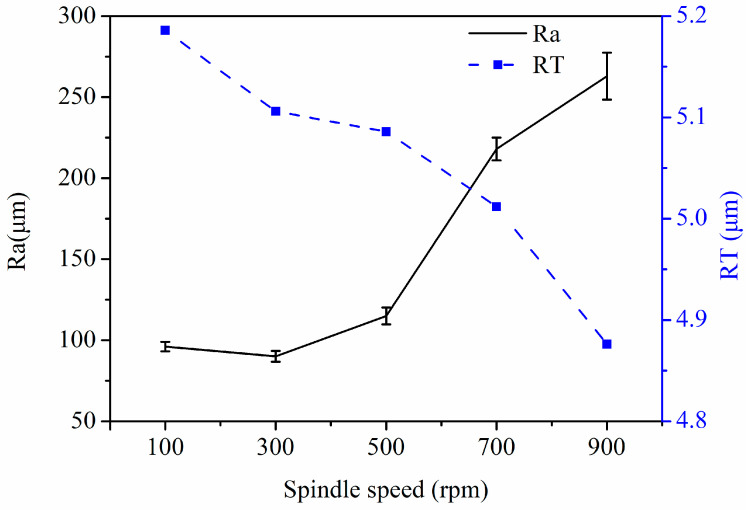
Graph of the effect of spindle speed on Ra and RT values.

**Figure 9 micromachines-14-00621-f009:**
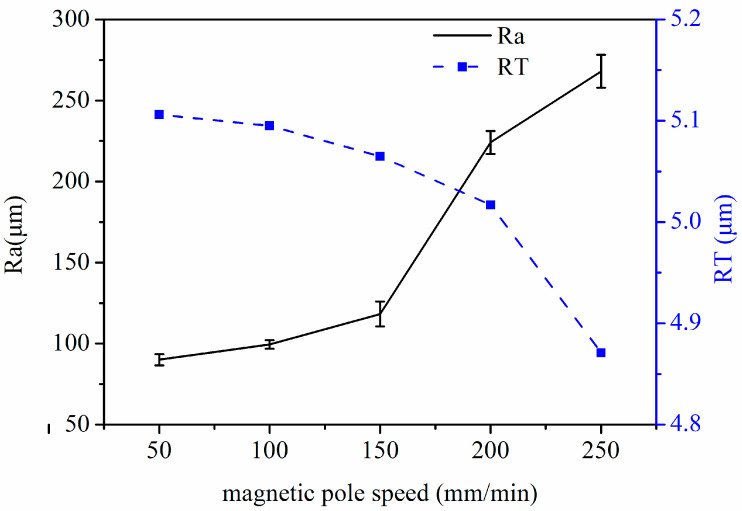
Graph of the effect of magnetic pole speed on Ra and RT values.

**Figure 10 micromachines-14-00621-f010:**
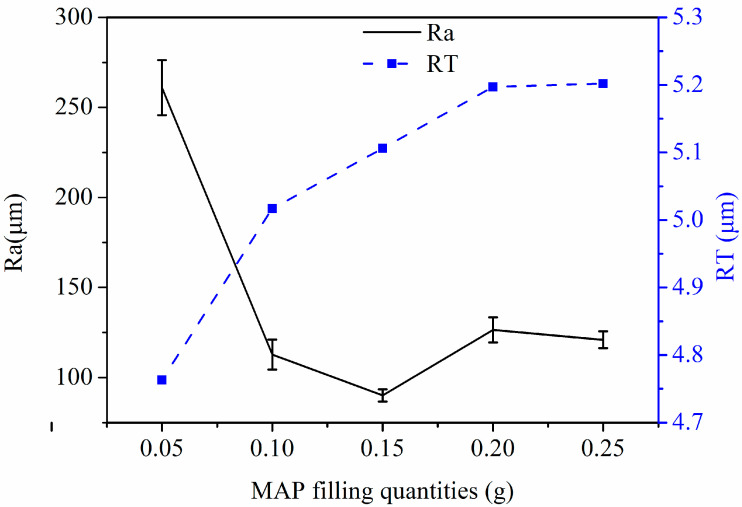
Graph of the effect of MAP filling quantities on Ra and RT values of tube inner wall.

**Figure 11 micromachines-14-00621-f011:**
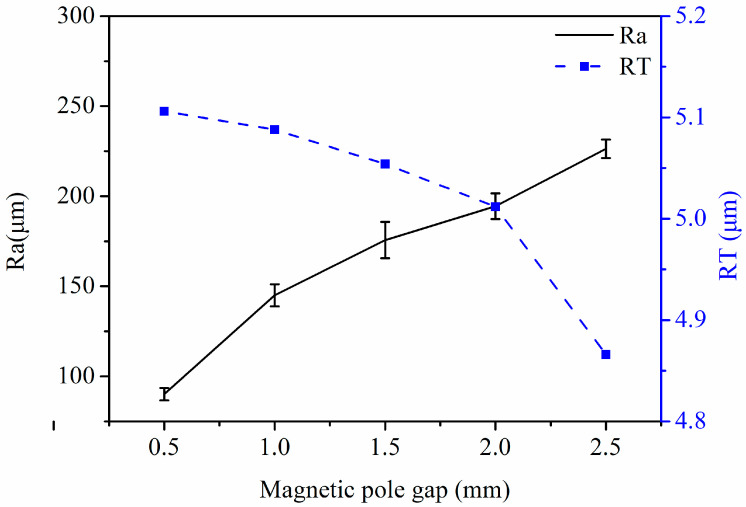
Graph of the effect of magnetic pole gap on Ra and RT of tube inner wall.

**Figure 12 micromachines-14-00621-f012:**
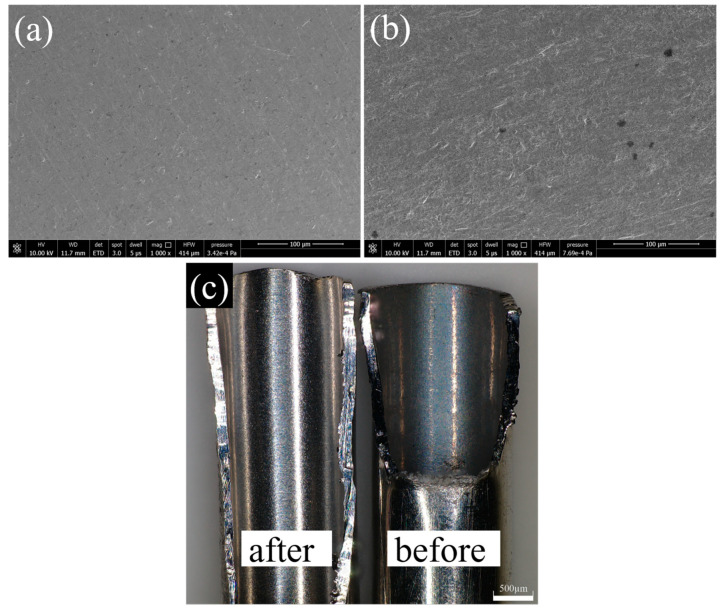
(**a**) SEM image of tube inner wall surface after finishing (Ra = 0.09 μm); (**b**) SEM image of tube inner wall surface before finishing (Ra = 0.337 μm); (**c**) comparison image of after and before finishing.

**Table 1 micromachines-14-00621-t001:** Preparation of raw materials and preparation parameters.

Raw Materials	Preparation Parameters
Spherical ferromagnetic metal powder (composition 99.9% Fe, particle size: 106~120 μm); alumina hard abrasive (d50 = 14 μm)	Ring seam nozzle (nozzle cone angle 65°, nozzle annular seam diameter 3.5 mm, nozzle bore diameter 46 mm, inlet pressure of nozzle 0.5 MPa); distance between nozzle and plasma generator 70 mm; I = 700 A, Ar = 1000 L/h, H_2_ = 200 L/h, iron powder 40 g/min, Al_2_O_3_ powder 240 g/min, equipment power 25.34 kW

**Table 2 micromachines-14-00621-t002:** Equipment performance parameters.

Parameter	Value
Spindle speed (rpm)	≤2000
Machining tube diameter (mm)	0.3–3
Machining tube length (mm)	100–2000
Magnetic pole speed (mm/min)	≤1000

**Table 3 micromachines-14-00621-t003:** Composition of Co–Cr cardiovascular stent tubes.

Element	Co	Cr	W	C	Ni	Mn	O
w/%	47–49	19–20	11–12	≤10	≤9	≤2	≤2

**Table 4 micromachines-14-00621-t004:** Mechanical properties of cobalt–chromium alloy cardiovascular stent tubes.

Performance Indicators	Density (g·cm^3^)	Modulus of Elasticity (GPa)	Tensile Strength (MPa)	Yield Strength	Elongation (%)
Value	9.2	243	820–1200	420–600	35–55

**Table 5 micromachines-14-00621-t005:** Experimental parameters.

Process Parameters	Values
Finishing time (h)	4
Al_2_O_3_ MAP particle size (μm)	125~150
Cutting fluid	V_Anti-rust emulsified oil_:V_Deionized water_ = 1:20
Spindle speed (rpm)	100, 300, 500, 700, 900
Magnetic pole speed (mm/min)	50, 100, 150, 200, 250
MAP loading quantities (g)	0.05, 0.10, 0.15, 0.20, 0.25
Magnetic pole gap (mm)	0.5, 1.0, 1.5, 2.0, 2.5

## Data Availability

Not applicable.

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
