# Peer review of "Investigation of Spherical Al2O3 Magnetic Abrasive Prepared by Novel Method for Finishing of the Inner Surface of Cobalt–Chromium Alloy Cardiovascular Stents Tube"

_micromachines, 2023, doi:10.3390/mi14030621_

Round 1
Reviewer 1 Report
The paper approach the problem of finishing with the help of abrasive magnetic powders.
The topic addressed is topical and of practical interest.
Numbers 6 to 9 are missing in the numbering of the pictures, while they are referred to in the text. The figures need to be renumbered.
The comment related to the current figure 12 states that the PSFI and RT decrease as the speed of the magnetic poles decreases. But the presented graph contradicts this conclusion. This situation must be corrected.
The heading of section 4.5 on page 11 has a portion that repeats “4.5. Results and discussions4.5Results and discussions ...”.
A similar situation is in the paragraph after the figure: "It can be observed from the figure that as the pole distance increases the PSFI and RT values of the inner wall decrease with the increase of the pole distance."
Figures a, b and c are referred to in the caption of figure 16, but are not marked in the figure. They should be clearly identified.
Author Response
Dear Reviewer:
I would first like to thank you for your letter and the reviewer’s comments concerning our manuscript entitled “Investigation of spherical Al2O3 magnetic abrasive prepared by novel method for finishing of the inner surface of cobalt-chromium alloy cardiovascular stents tube” The comments you made were all valuable and helpful for revising and improving our paper and the important guiding significance to our research. We have substantially revised our manuscript after reading the comments provided by the reviewer. I hope to be met with approval.
Revised portions are marked in red throughout the paper, and the main corrections in the paper and the responses to the reviewer’s comments are as follows:
Responds to the reviewer’s comments:
Reviewer 1:
1) Numbers 6 to 9 are missing in the numbering of the pictures, while they are referred to in the text. The figures need to be renumbered.
Answer: We are sorry for our negligence. I have revised my manuscript, and the specific revised parts have been marked in red in the submitted manuscript.
2) The comment related to the current figure 12 states that the PSFI and RT decrease as the speed of the magnetic poles decreases. But the presented graph contradicts this conclusion. This situation must be corrected.
Answer: We are very sorry for our negligence of this section. I have revised my manuscript, and the specific revised parts have been marked in red in the submitted manuscript.
3) The heading of section 4.5 on page 11 has a portion that repeats “4.5. Results and discussions4.5Results and discussions ...”.
Answer: We are very sorry for our negligence. I have revised my manuscript, and the specific revised parts have been marked in red in the submitted manuscript.
4) A similar situation is in the paragraph after the figure: "It can be observed from the figure that as the pole distance increases the PSFI and RT values of the inner wall decrease with the increase of the pole distance."
Answer: We are very sorry for our negligence of this section. I have revised my manuscript, and the specific revised parts have been marked in red in the submitted manuscript.
5)Figures a, b and c are referred to in the caption of figure 16, but are not marked in the figure. They should be clearly identified.
Answer: We are very sorry for our negligence. I have revised my manuscript, and marked in the figure.
We tried our best to improve the revised manuscript and made some changes in the revised manuscript. And here we did not list the changes but marked in red in the revised manuscript, using the "Track Changes" function in Microsoft Word.
We appreciate Editors/Reviewer’s warm work earnestly, and hope that the correction will meet with approval. Once more, thank you very much for your comments and suggestions.
Thanks very much again for your attention to our manuscript. Once again, thank you for your help to our manuscript processing.
Yours sincerely,
Yugang Zhao

Reviewer 2 Report
The manuscript “Investigation of spherical Al2O3 magnetic abrasive prepared by novel method for finishing of the inner surface of cobalt-chromium alloy cardiovascular stents tube” discussed a novel process of polishing inner surface of cardiovascular stents tubes. Here are the reasons why the current manuscript cannot be accepted to Micromachines.
1. For the data used in the Tables and Figures in the manuscript, the authors failed to mention where they are coming from or how they are acquired. Examples:
a) In Table 2, there is no EDS picture evidence showing the atomic fraction or the element distribution across the powder.
b) In Table 3 (page 8) and Table 4, it is not clear if the tube used is widely commercially available or it is customized. It is necessary to let the readers know how the data are acquired.
c) Evidence of the roughness measurements from the microscope are needed for Figures 11-15.
2. The indices in the manuscript are very confusing. Examples:
a) There are two Table 3s.
b) In Results and Discussions, the contexts and the indices of the figures are all off.
c) Conclusion (3) is missing.
3. The results are not conclusive. Examples:
a) In Figure 16, it is not clear about the change in the before and after images, especially in the comparison image in (c).
b) The authors did the experiments on five process parameters, but the analysis of the optimization is lacking.
c) The authors claimed in conclusion (4) that “surface defects such as folds, cracks, scratches and pits generated during the production process” can be removed, but there is no evidence of the removal of “folds”, “cracks” or “scratches”, at least not in Fig. 16.
4. Figure 2 reuses the schematic from another paper (Preparation of Al2O3 magnetic abrasives by combining plasma molten metal powder with sprayed abrasive powder) without proper permission granted.
Author Response
Dear Reviewer:
I would first like to thank you for your letter and the reviewer’s comments concerning our manuscript entitled “Investigation of spherical Al2O3 magnetic abrasive prepared by novel method for finishing of the inner surface of cobalt-chromium alloy cardiovascular stents tube” The comments you made were all valuable and helpful for revising and improving our paper and the important guiding significance to our research. We have substantially revised our manuscript after reading the comments provided by the reviewer. I hope to be met with approval.
Revised portions are marked in red throughout the paper, and the main corrections in the paper and the responses to the reviewer’s comments are as follows:
Responds to the reviewer’s comments:
Reviewer 2:
- For the data used in the Tables and Figures in the manuscript, the authors failed to mention where they are coming from or how they are acquired. Examples:
- a) In Table 2, there is no EDS picture evidence showing the atomic fraction or the element distribution across the powder.
Answer: We are sorry for our negligence. I have revised my manuscript, and delete this part. The specific revised parts have been marked in red in the submitted manuscript.
- b) In Table 3 (page 8) and Table 4, it is not clear if the tube used is widely commercially available or it is customized. It is necessary to let the readers know how the data are acquired.
Answer: We are sorry for our negligence. I have revised my manuscript, the specific revised parts have been marked in red in the submitted manuscript. The tube used is widely commercially available, donated by Shenzhen Kinhely Biotechnology Co., Ltd.
- c) Evidence of the roughness measurements from the microscope are needed for Figures 11-15.
Answer: Thanks for the suggestion of the reviewer. I have revised my manuscript, and The PSFI value was modified to the roughness of surface and the corresponding picture was modified. The specific revised parts have been marked in red in the submitted manuscript.
- The indices in the manuscript are very confusing. Examples:
- a) There are two Table 3s.
Answer: We are very sorry for our negligence. I have revised my manuscript, and the specific revised parts have been marked in red in the submitted manuscript.
- b) In Results and Discussions, the contexts and the indices of the figures are all off.
Answer: We are very sorry for our negligence. I have revised my manuscript, and the specific revised parts have been marked in red in the submitted manuscript.
- c) Conclusion (3) is missing.
Answer: We are very sorry for our negligence. I have revised my manuscript, and the specific revised parts have been marked in red in the submitted manuscript.
- The results are not conclusive. Examples:
- a) In Figure 16, it is not clear about the change in the before and after images, especially in the comparison image in (c).
Answer: We are very sorry for our negligence. I have revised my manuscript, and annotated in the images. The specific revised parts have been marked in red in the submitted manuscript.
- b) The authors did the experiments on five process parameters, but the analysis of the optimization is lacking.
Answer: Thanks for the suggestion of the reviewer. The focus of this paper is the polishing and defect removal of the inner wall of cardiovascular stent tubing using Al2O3 magnetic abrasive prepared by the new method. The study on parameter optimization is expected to appear in the next paper and is ready to be submitted to Micromachines.
- c) The authors claimed in conclusion (4) that “surface defects such as folds, cracks, scratches and pits generated during the production process” can be removed, but there is no evidence of the removal of “folds”, “cracks” or “scratches”, at least not in Fig. 16.
Answer: Thanks for the suggestion of the reviewer. I have revised my manuscript, the specific revised parts have been marked in red in the submitted manuscript.
- Figure 2 reuses the schematic from another paper (Preparation of Al2O3 magnetic abrasives by combining plasma molten metal powder with sprayed abrasive powder) without proper permission granted.
Answer: We are very sorry for our negligence. The copyright file was sent to the Micromachines editorial office in a timely manner
We tried our best to improve the revised manuscript and made some changes in the revised manuscript. And here we did not list the changes but marked in red in the revised manuscript, using the "Track Changes" function in Microsoft Word.
We appreciate Editors/Reviewer’s warm work earnestly, and hope that the correction will meet with approval. Once more, thank you very much for your comments and suggestions.
Thanks very much again for your attention to our manuscript. Once again, thank you for your help to our manuscript processing.
Yours sincerely,
Yugang Zhao

Reviewer 3 Report
This article presents research related to the finishing of the inner surface of cardiovascular stent tubes. The subject matter is important and topical. The medical industry is one of the more intensively growing industries. According to WHO cardiovascular diseases are the leading cause of death globally. The method used was magnetic abrasive finishing. The abrasive material used was Al2O3 combined with spherical metal powder.
Despite the existence of research, the article is missing some key information. My comments are as follows:
1. The authors present the MEF method as if it were being presented for the first time in the literature. Meanwhile, at least two articles (developed by a similar team) have already been published that present this method:
Song, Z.; Zhao, Y.; Li, Z.; Cao, C.; Liu, G.; Liu, Q.; Zhang, X.; Dai, D.; Zheng, Z.; Zhao, C.; et al. Study on the Micro Removal Process of Inner Surface of Cobalt Chromium Alloy Cardiovascular Stent Tubes. Micromachines 2022, 13, 1374. https://doi.org/10.3390/ mi13091374
Deng, Y., Zhao, Y., Zhao, G. et al. Study on magnetic abrasive finishing of the inner surface of Ni–Ti alloy cardiovascular stents tube. Int J Adv Manuf Technol 118, 2299–2309 (2022). https://doi.org/10.1007/s00170-021-08074-3
Not citing these works seems a bit odd. I am not diminishing the work that has been made. However, the article should clearly state what is a 'new' contribution. In the description of the results, it is also worthwhile to have a discussion and refer to previous research results - performed with other abrasives and/or machining parameters.
2. In section 3.4, the parameters for measuring and filtering the surface roughness should be provided.
3. In Table 5 you can see what values the various process parameters took. However, the tests carried out were single-factor tests - as we read in the summary and as can be seen from the graphs. This, however, should be clearly pointed out in section 3.4. It is also not indicated what values the fixed parameters took in a given test. E.g. when testing spindle speed it took on values of 100, 300, 500, 700 and 900 rpm, but what were: MAPs filling quantity, particle size, and magnetic poles speed?
4. The section titles in chapter 4 are too long. E.g. instead of "Results and discussion Results and discussion of the effect of spindle speed on PSFI and RT of the inner wall of the tube", "The effect of spindle speed on PSFI and RT" would be sufficient.
5. What is Ff force component?
6. The results should indicate at least limits the roughness changed before and after polishing.
7. In the current Figure 16, figures a, b and c need to be labeled. It is also not clear which surface is before and after polishing.
8. In conclusion, write about the results of current research. Do not present previously presented methods as a new contribution.
9. In the first sentence talking about the state of 'today' there is a reference to two publications from 2019 - more up-to-date data can be quoted, e.g. from the HWO website.
10. Introduction is a bit hard to read - the text should be split into more paragraphs.
11. What is the efficiency of combining plasma molten metal powder with sprayed abrasive powder?
12. The information provided in table 1 is unclear - especially as regards Preparation parameters. All symbols in the table should be explained.
13. Why in the article particle sizes are given with a minus sign?
14. Figure numbers - in captions and in references are incorrect.
15. In Table 3, the headings should be Parameter (rather than Performance) and Value (rather than Parameter).
Author Response
Dear Reviewer:
I would first like to thank you for your letter and the reviewer’s comments concerning our manuscript entitled “Investigation of spherical Al2O3 magnetic abrasive prepared by novel method for finishing of the inner surface of cobalt-chromium alloy cardiovascular stents tube” The comments you made were all valuable and helpful for revising and improving our paper and the important guiding significance to our research. We have substantially revised our manuscript after reading the comments provided by the reviewer. I hope to be met with approval.
Revised portions are marked in red throughout the paper, and the main corrections in the paper and the responses to the reviewer’s comments are as follows:
Responds to the reviewer’s comments:
Reviewer 3:
- The authors present the MEF method as if it were being presented for the first time in the literature. Meanwhile, at least two articles (developed by a similar team) have already been published that present this method:
Song, Z.; Zhao, Y.; Li, Z.; Cao, C.; Liu, G.; Liu, Q.; Zhang, X.; Dai, D.; Zheng, Z.; Zhao, C.; et al. Study on the Micro Removal Process of Inner Surface of Cobalt Chromium Alloy Cardiovascular Stent Tubes. Micromachines 2022, 13, 1374. https://doi.org/10.3390/ mi13091374
Deng, Y., Zhao, Y., Zhao, G. et al. Study on magnetic abrasive finishing of the inner surface of Ni–Ti alloy cardiovascular stents tube. Int J Adv Manuf Technol 118, 2299–2309 (2022). https://doi.org/10.1007/s00170-021-08074-3
Not citing these works seems a bit odd. I am not diminishing the work that has been made. However, the article should clearly state what is a 'new' contribution. In the description of the results, it is also worthwhile to have a discussion and refer to previous research results - performed with other abrasives and/or machining parameters.
Answer: Thank you for your comments on this part. After careful consideration, I have decided to add both references and revise this part from the article, and the specific revised parts have been marked in red in the submitted manuscript.
- In section 3.4, the parameters for measuring and filtering the surface roughness should be provided.
Answer: Thank you for your comments on this section. I have revised my manuscript, and the specific revised parts have been marked in red in the submitted manuscript.
- In Table 5 you can see what values the various process parameters took. However, the tests carried out were single-factor tests - as we read in the summary and as can be seen from the graphs. This, however, should be clearly pointed out in section 3.4. It is also not indicated what values the fixed parameters took in a given test. E.g. when testing spindle speed it took on values of 100, 300, 500, 700 and 900 rpm, but what were: MAPs filling quantity, particle size, and magnetic poles speed?
Answer: We are sorry for our negligence. I have revised my manuscript, and the specific revised parts have been marked in red in the submitted manuscript.
- The section titles in chapter 4 are too long. E.g. instead of "Results and discussion Results and discussion of the effect of spindle speed on PSFI and RT of the inner wall of the tube", "The effect of spindle speed on PSFI and RT" would be sufficient.
Answer: We are sorry for our negligence. I have revised my manuscript, and the specific revised parts have been marked in red in the submitted manuscript.
- What is Ff force component?
Answer: We are sorry for our negligence. I have revised my manuscript, and the specific revised parts have been marked in red in the submitted manuscript.
- The results should indicate at least limits the roughness changed before and after polishing.
Answer: Thank you for your comments on this part. I have revised my manuscript, and the specific revised parts have been marked in red in the submitted manuscript.
- In the current Figure 16, figures a, b and c need to be labeled. It is also not clear which surface is before and after polishing.
Answer: We are sorry for our negligence. I have revised my manuscript, and modify and annotate the corresponding figure. The specific revised parts have been marked in red in the submitted manuscript.
- In conclusion, write about the results of current research. Do not present previously presented methods as a new contribution.
Answer: Thank you for your comments on this part. I have revised my manuscript, and the specific revised parts have been marked in red in the submitted manuscript.
9.In the first sentence talking about the state of 'today' there is a reference to two publications from 2019 - more up-to-date data can be quoted, e.g. from the HWO website.
Answer: Thanks for the suggestion of the reviewer. I have revised my manuscript, and the specific revised parts have been marked in red in the submitted manuscript.
- Introduction is a bit hard to read - the text should be split into more paragraphs.
Answer: Thank you for your comments on this part. I have revised my manuscript, and the specific revised parts have been marked in red in the submitted manuscript.
- What is the efficiency of combining plasma molten metal powder with sprayed abrasive powder?
Answer: Thanks for the suggestion of the reviewer. The efficiency of combining plasma molten metal powder with sprayed abrasive powder is exceed 40g/min. I have revised my manuscript, and the specific revised parts have been marked in red in the submitted manuscript.
- The information provided in table 1 is unclear - especially as regards Preparation parameters. All symbols in the table should be explained.
Answer: We are very sorry for our negligence of this section. I have revised my manuscript, and the specific revised parts have been marked in red in the submitted manuscript.
- Why in the article particle sizes are given with a minus sign?
Answer: We are very sorry for our negligence. I have revised my manuscript, and the specific revised parts have been marked in red in the submitted manuscript.
- Figure numbers - in captions and in references are incorrect.
Answer: We are very sorry for our negligence of this section. I have revised my manuscript, and the specific revised parts have been marked in red in the submitted manuscript.
- In Table 3, the headings should be Parameter (rather than Performance) and Value (rather than Parameter).
Answer: Thanks for the comments and suggestion of the reviewer. We should make an apology for the failure to clearly interpret. I have revised my manuscript, and the specific revised parts have been marked in red in the submitted manuscript.
We tried our best to improve the revised manuscript and made some changes in the revised manuscript. And here we did not list the changes but marked in red in the revised manuscript, using the "Track Changes" function in Microsoft Word.
We appreciate Editors/Reviewer’s warm work earnestly, and hope that the correction will meet with approval. Once more, thank you very much for your comments and suggestions.
Thanks very much again for your attention to our manuscript. Once again, thank you for your help to our manuscript processing.
Yours sincerely,
Yugang Zhao

Round 2
Reviewer 2 Report
The manuscript “Investigation of spherical Al2O3 magnetic abrasive prepared by novel method for finishing of the inner surface of cobalt-chromium alloy cardiovascular stents tube” discussed a novel process of polishing inner surface of cardiovascular stents tubes. The authors have fixed some big issues (including the plagiarism of Figure 2, which didn’t reflect on the manuscript) in the first draft and could be accepted. Small errors need to be fixed before publication. For example, proper spaces are needed in between “andRT” in the abstract; two fonts of words are in the abstract, etc.
Author Response
Dear Reviewer:
I would first like to thank you for your letter and the reviewer’s comments concerning our manuscript entitled “Investigation of spherical Al2O3 magnetic abrasive prepared by novel method for finishing of the inner surface of cobalt-chromium alloy cardiovascular stents tube” The comments you made were all valuable and helpful for revising and improving our paper and the important guiding significance to our research. We have substantially revised our manuscript after reading the comments provided by the reviewer. I hope to be met with approval.
Revised portions are marked in red throughout the paper, and the main corrections in the paper and the responses to the reviewer’s comments are as follows:
Responds to the reviewer’s comments:
Reviewer 2:
- The manuscript “Investigation of spherical Al2O3 magnetic abrasive prepared by novel method for finishing of the inner surface of cobalt-chromium alloy cardiovascular stents tube” discussed a novel process of polishing inner surface of cardiovascular stents tubes. The authors have fixed some big issues (including the plagiarism of Figure 2, which didn’t reflect on the manuscript) in the first draft and could be accepted. Small errors need to be fixed before publication. For example, proper spaces are needed in between “andRT” in the abstract; two fonts of words are in the abstract, etc.
Answer: We are very sorry for our negligence. We are sorry for our negligence. I have revised my manuscript, the specific revised parts have been marked in red in the submitted manuscript. The copyright file is sent to the editor by email.
We tried our best to improve the revised manuscript and made some changes in the revised manuscript. And here we did not list the changes but marked in red in the revised manuscript, using the "Track Changes" function in Microsoft Word.
We appreciate Editors/Reviewer’s warm work earnestly, and hope that the correction will meet with approval. Once more, thank you very much for your comments and suggestions.
Thanks very much again for your attention to our manuscript. Once again, thank you for your help to our manuscript processing.
Yours sincerely,
Yugang Zhao

Reviewer 3 Report
Thank you for responding to all the comments. The authors have responded to most of them and made appropriate corrections. However, in section 3.4, the parameters for measuring and filtering the surface roughness are still missing.
Author Response
Dear Reviewer:
I would first like to thank you for your letter and the reviewer’s comments concerning our manuscript entitled “Investigation of spherical Al2O3 magnetic abrasive prepared by novel method for finishing of the inner surface of cobalt-chromium alloy cardiovascular stents tube” The comments you made were all valuable and helpful for revising and improving our paper and the important guiding significance to our research. We have substantially revised our manuscript after reading the comments provided by the reviewer. I hope to be met with approval.
Revised portions are marked in red throughout the paper, and the main corrections in the paper and the responses to the reviewer’s comments are as follows:
Responds to the reviewer’s comments:
Reviewer 3:
1.Thank you for responding to all the comments. The authors have responded to most of them and made appropriate corrections. However, in section 3.4, the parameters for measuring and filtering the surface roughness are still missing.
Answer: We are very sorry for our negligence. I have revised my manuscript, the specific revised parts have been marked in red in the submitted manuscript. The copyright file is uploaded as an attachment.
We tried our best to improve the revised manuscript and made some changes in the revised manuscript. And here we did not list the changes but marked in red in the revised manuscript, using the "Track Changes" function in Microsoft Word.
We appreciate Editors/Reviewer’s warm work earnestly, and hope that the correction will meet with approval. Once more, thank you very much for your comments and suggestions.
Thanks very much again for your attention to our manuscript. Once again, thank you for your help to our manuscript processing.
Yours sincerely,
Yugang Zhao
